# Designing Bivariate Auto-Regressive Timeseries with Controlled Granger Causality

**DOI:** 10.3390/e23060742

**Published:** 2021-06-12

**Authors:** Shohei Hidaka, Takuma Torii

**Affiliations:** Japan Advanced Institute of Science and Technology, 1-1 Asahidai, Nomi 923-1292, Ishikawa, Japan; tak.torii@jaist.ac.jp

**Keywords:** Granger causality, transfer entropy, vector auto-regressive model, Lyapunov equation

## Abstract

In this manuscript, we analyze a bivariate vector auto-regressive (VAR) model in order to draw the design principle of a timeseries with a controlled statistical inter-relationship. We show how to generate bivariate timeseries with given covariance and Granger causality (or, equivalently, transfer entropy), and show the trade-off relationship between these two types of statistical interaction. In principle, covariance and Granger causality are independently controllable, but the feasible ranges of their values which allow the VAR to be proper and have a stationary distribution are constrained by each other. Thus, our analysis identifies the essential tri-lemma structure among the stability and properness of VAR, the controllability of covariance, and that of Granger causality.

## 1. Introduction

### 1.1. Background and Motivation

In the field of cognitive psychology, the human perception of the life-likeness (called *animacy perception*) of one or multiple moving geometric patterns has been studied for decades [1,2,3,4,5]. There are multiple findings on the effect of “synchrony” or “temporal contingency” between multiple moving points on animacy perception. Findings from one line of research [2] have suggested that a higher degree of “temporal contingency” of the moving objects is related to a higher likelihood of animacy perception. Findings from the other line of research [6] have suggested that the highest “temporal contingency”, presented in the form of perfect synchronization, would decrease the likelihood of animacy perception.

These two lines of research have together suggested the existence of multiple types of “temporal contingency”. Nevertheless, this past research does not appear to clarify these types. Further, confusion surrounding these two distinct types of effects have led to two lines of apparently conflicting effects of “temporal contingency”.

With this potential conflict in the literature on animacy perception in mind, we explore a theoretical framework which can generate timeseries of multiple random variables with multiple distinct types of statistical dependency. One such system, which is sufficiently simple and readily manipulable, is vector auto-regression (VAR). Vector auto-regression is a random process for generating multivariate timeseries for a given set of parameters. In this manuscript, we specifically consider only bivariate VAR, which is a minimal system with interaction between two moving points.

### 1.2. Vector Auto-Regressive Model, Granger Causality, and Transfer Entropy

Importantly, bivariate VAR, a series of paired random variables $\left(x_{t},y_{t}\right)$ for t=0,1,…, has two types of statistical dependency—that is, the correlation and *Granger causality* of a timeseries [7], which has been identified as transfer entropy [8] of the timeseries generated by a Gaussian process by [9]. The correlation between univariate series *x* and *y* is a statistical dependency between $x_{t}$ and $y_{t}$ in the limit t→∞ (if it exists), while a Granger causality from *y* to *x* is that between $x_{t}$ and $y_{t-1}$ given $x_{t-1}$ in the limit t→∞ (if it exists). Conceptually, correlation captures a type of similarity between two timeseries, whereas Granger causality captures the “reactiveness” of one timeseries to another.

Thus, given these differences in theory, our goal is to propose a theoretical method to generate a bivariate random timeseries with a desired correlation and two types. The Granger causality of VAR has also been considered in other fields. In econometrics, the field in which it was originally proposed [7], Granger causality has been used as a measure of interaction between a pair of economic timeseries [10,11,12]. It has also been used in general behavioral sciences [13,14], particularly in computational neuroscience [15,16,17,18]. Such an in-principle data generation technique would be vital to the testing of any hypothesis on the empirical nature of timeseries (e.g., animacy perception) in the empirical sciences using VAR and Granger causality, as mentioned above. To our knowledge, however, there has been no mathematical analysis of the theoretical limitation of such a data generation technique for a given statistics.

Thus, we first need to explore the mathematical relationship among the parameters in a VAR with the correlation and Granger causality in a timeseries generated from it. In this paper, we therefore explore a theoretical structure of bivariate VAR from the designer’s perspective, and analyze a mathematical limit to the extent which we can simultaneously control correlation and Granger causality of a bivariate timeseries.

This paper is written with the following structure. In Section 2, the VAR model is defined, from which a set of basic statistical properties of the VAR model are derived, such as Granger causality (Section 3) with a set of parameters. In Section 4, the existence of the stationary distribution of the VAR is analyzed. This is a foundation which sets the limit of a controllable set of parameters. In Section 5, we give a method to derive the parameters of VAR for a given set of statistics in bivariate timeseries. In Section 6, the mathematical analysis provided in this paper is summarized and a remark on the design principle of bivariate timeseries generated by VAR is added.

## 2. Vector Auto-Regression (VAR)

In theory, Granger causality (GC) is the transfer entropy of random variables in a bivariate vector auto-regression (VAR) model up to a constant factor of 2, if the VAR model has a stationary distribution [9]. Thus, it is straightforward to start with the bivariate VAR and derive its transfer entropy. In this way, we can derive a rich mathematical relationship between GC and the properties of VAR, rather than just a statistics of the bivariate timeseries.

**Definition** **1.**
*For some real vector $\mu \in R^{2}$ and some positive-definite matrix $\Sigma \in R^{2\times 2}$, suppose the random variable $\epsilon_{t}$ for every integer t=0,1,… is drawn from the bivariate normal distribution $N(\epsilon_{t}|\mu ,\Sigma )$ with mean μ and variance *Σ*. Define the initial vector by $v_{0}=\left(\begin{array}{c} x_{0}\\ y_{0} \end{array}\right)$, and for t≥0 and a given coefficient matrix with real entries $A:=\left(\begin{array}{cc} a_{0,0} & a_{0,1}\\ a_{1,0} & a_{1,1} \end{array}\right)\in R^{2\times 2}$, define the random variable $v_{t}$ by*
(1)$$v_{t+1}:=Av_{t}+\epsilon_{t}.$$
*Then, bivariate vector auto-regression is defined by the semi-infinite series of random variables $V=(v_{0},v_{1},v_{2},\ldots )$.*


In general, one can generate a timeseries $v_{0},v_{1},\ldots$ by fixing a set of the VAR parameters, the coefficient matrix *A*, and the base covariance matrix Σ, where the base mean vector μ is omitted as its effect is lost in the limit t→∞ when the VAR is stationary. The stationary correlation (covariance) $\hat{\Sigma }$ and Granger causality $G_{0}$ and $G_{1}$ defined later are the statistics of the timeseries generated by a VAR model (Figure 1). In what follows, we first explore the forward relationship of how the statistics $\hat{\Sigma }$ and $G_{0},G_{1}$ are given by the VAR parameters (A,Σ). We then consider the backward relationship in which the VAR parameters (A,Σ) suffice to generate a timeseries with given desired timeseries statistics $\left(G_{0},G_{1},\hat{\Sigma }\right)$.

### 2.1. Marginal Distribution of the VAR at Each Step

**Lemma** **1**(Marginal distribution of the VAR random variables at each step)**.**
*The VAR model with the initial vector $v_{0}\in R^{2}$ and the coefficient matrix $A\in R^{2\times 2}$ has the bivariate normal distribution*
$$N(v_{t}|\mu_{t},\Sigma_{t})$$
*as its marginal distribution of the random variable $v_{t}$ at each step t=0,1,…, where*
$$\mu_{t}:=A^{t}v_{0},\;\Sigma_{t}=\sum_{s=0}^{t}A^{s}\Sigma {\left(A^{s}\right)}^{⊤}.$$

**Proof.** By Definition 1, Lemma 1 holds for t=0. For t+1>0, we prove Lemma 1 by assuming that it holds up to t≥0. By this assumption held for *t*, we have the distribution of $v_{t}\in R^{2}$ as the bivariate normal distribution
$$N\left(v_{t}|\mu_{t},\Sigma_{t}\right)={\left(2\pi \right)}^{-1}{\left|\Sigma_{t}\right|}^{-\frac{1}{2}}e^{-\frac{1}{2}{\left(v_{t}-\mu_{t}\right)}^{⊤}\Sigma_{t}^{-1}\left(v_{t}-\mu_{t}\right)}$$
with its mean $\mu_{t}$ and its covariance matrix $\Sigma_{t}$. Then, the random variable $Av_{t}$ is distributed by the normal distribution
(2)$$\begin{array}{ccc} N(Av_{t}|A\mu_{t},A\Sigma_{t}A^{⊤}) & = & {\left(2\pi \right)}^{-1}{\left|A\Sigma_{t}A^{⊤}\right|}^{-\frac{1}{2}}\\ & \times & e^{-\frac{1}{2}{\left(A\left(v_{t}-\mu_{t}\right)\right)}^{⊤}{\left(A\Sigma_{t}A^{⊤}\right)}^{-1}\left(A\left(v_{t}-\mu_{t}\right)\right)} \end{array}$$
with its mean $A\mu_{t}$ and the covariance matrix $A\Sigma_{t}A^{⊤}$. The random variable $\epsilon_{t}$ is distributed by the following normal distribution:
$$N\left(\epsilon_{t}|0,\Sigma \right)={\left(2\pi \right)}^{-1}{\left|\Sigma \right|}^{-\frac{1}{2}}e^{-\frac{1}{2}\epsilon_{t}^{⊤}\Sigma^{-1}\epsilon_{t}}.$$
Thus, by VAR, Equation (Equation 1), we have the random variable
$$v_{t+1}:=Av_{t}+\epsilon_{t}$$
which has a distribution calculated by the following integral:
$$P\left(v_{t+1}\right)=\int_{\epsilon_{t}\in R^{2}}N\left(v_{t+1}-\epsilon_{t}|A\mu_{t},A\Sigma_{t}A^{⊤}\right)N\left(\epsilon_{t}|0,\Sigma \right)d\epsilon_{t}.$$
Calculating this, we have
(3)$$P\left(v_{t+1}\right)=N\left(v_{t+1}|A\mu_{t},A\Sigma_{t}A^{⊤}+\Sigma \right).$$
Thus defining by $\mu_{t+1}:=A\mu_{t}$ and $\Sigma_{t+1}=A\Sigma_{t}A^{⊤}+\Sigma$, Lemma 1 holds for t+1. By expanding this, we have the Lemma 1 for any integer t≥0. □

### 2.2. Stability of VAR: Lyapunov Equation

By Lemma 1, the mean and covariance matrix of the random variable at the tth step are
$$\mu_{t}=A^{t}v_{0}\;\mathrm{and}\;\Sigma_{t}=\sum_{s=0}^{t}A^{s}\Sigma {\left(A^{s}\right)}^{⊤}.$$
From this, we have the stationary distribution
$$\lim_{t\to \infty }N\left(v_{t}|\mu_{t},\Sigma_{t}\right),$$
if and only if the absolute values of all the eigenvalues $\lambda_{0},\lambda_{1}\in C$ of the coefficient matrix *A* are less than 1. If there is such a stationary distribution, we call the VAR *stable*, and its stationary distribution is the bivariate normal distribution
$$N(\hat{v}|0_{2},\hat{\Sigma }),$$
where the stationary mean vector $\hat{v}\in R^{2}$ and stationary covariance matrix $\hat{\Sigma }\in R^{2\times 2}$ are defined as follows. If the VAR is stable, we have the following Lyapunov equation of the stationary covariance matrix $\hat{\Sigma }\in R^{2\times 2}$:(4)$$\hat{\Sigma }=\Sigma +A\hat{\Sigma }A^{⊤}.$$
The Lyapunov equation is solved analytically by
(5)$$\mathrm{vec}\left(\hat{\Sigma }\right)={\left(I_{4}-A⊗A\right)}^{-1}\mathrm{vec}\left(\Sigma \right),$$
where $I_{d}\in R^{d\times d}$ is the dth order identity matrix, ⊗ denotes the Kronecker product, and $\mathrm{vec}\left(X\right)$ for any matrix $X={\left(x_{i,j}\right)}_{i=1,\ldots ,n,j=1,\ldots ,m}$ is the vectorization operator $\mathrm{vec}():R^{n\times m}\to R^{nm\times 1}$ defined by
$$\mathrm{vec}\left(X\right):={\left(x_{1,1},x_{2,1},\ldots ,x_{n,1},\ldots ,x_{1,m},x_{2,m},\ldots ,x_{n,m}\right)}^{⊤}.$$

The Lyapunov Equation (Equation 4) has the solution for $\hat{\Sigma }$ if the VAR is stable, but not vice versa. This is shown by Lemma 4.

## 3. Transfer Entropy and Granger Causality

In [9], the transfer entropy of an appropriate triplet of variables in the VAR model is shown to be equivalent to Granger causality up to the constant factor 2. Following this guide, we define this quantity as the Granger causality of the VAR model, below.

Although this relationship has been known in a more general form [9], we re-derive it for bivariate VAR in order to later analyze the structure of VAR and GCs in depth—for example, its upper and lower bounds (Lemma 3), stability (Section 4), and design principle (Section 5).

**Definition** **2.**
*If VAR with its random variables $v_{t}={\left(x_{t},y_{t}\right)}^{⊤}\in R^{2}$ for t=0,1,… is stable, transfer entropy from y to x is defined by*
$$T_{y\to x}:=\lim_{t\to \infty }\left(H\left(x_{t+1}|x_{t}\right)-H\left(x_{t+1}|x_{t},y_{t}\right)\right),$$
*and the transfer entropy from x to y is defined by*
$$T_{x\to y}:=\lim_{t\to \infty }\left(H\left(y_{t+1}|y_{t}\right)-H\left(y_{t+1}|y_{t},x_{t}\right)\right),$$
*where the differential entropy of random variable x with its probability density function P is*
$$H\left(x\right):=-\int_{x\in \Omega }P\left(x\right)\log P\left(x\right)dx,$$
*and the conditional entropy is*
H(x|y):=H(x,y)−H(y).
*In particular, we call two times of transfer entropy Granger causality denoted by*
(6)$$G_{0}=2T_{y\to x}\;\mathrm{and}\;G_{1}=2T_{x\to y}.$$


Specifically, GCs are specifically written by the terms of the VAR parameters in the following lemma.

**Lemma** **2**(Granger causality)**.**
*If a stable VAR has its covariance matrix, coefficient matrix, and stationary matrix*
$$\Sigma =\left(\begin{array}{cc} \sigma_{0,0} & \sigma_{0,1}\\ \sigma_{1,0} & \sigma_{1,1} \end{array}\right),\;A=\left(\begin{array}{cc} a_{0,0} & a_{0,1}\\ a_{1,0} & a_{1,1} \end{array}\right),\;\hat{\Sigma }=\left(\begin{array}{cc} {\hat{\sigma }}_{0,0} & {\hat{\sigma }}_{0,1}\\ {\hat{\sigma }}_{1,0} & {\hat{\sigma }}_{1,1} \end{array}\right),$$
*each Granger causality of this VAR for i=0,1 is*
(7)$$\begin{array}{ccc} G_{i} & = & \log\left(1+\frac{a_{i,1-i}^{2}\det\left(\hat{\Sigma }\right)}{{\hat{\sigma }}_{i,i}\sigma_{i,i}}\right). \end{array}$$

**Proof.** In general, the differential entropy of multivariate normal distribution N(v|μ,Σ) is
$$H\left(v\right)=\frac{1}{2}\log\left|2\pi e\Sigma \right|,$$
where e≈2.71 is Napier’s constant. For the joint probability distribution of $v_{t}={\left(x_{t},y_{t}\right)}^{⊤}$
$$P\left(v_{t+1}|v_{t}\right)=N\left(v_{t+1}|Av_{t},\Sigma \right),$$
the two marginal probability distributions of $x_{t},y_{t}$ are
$$P\left(x_{t+1}|v_{t}\right)=N\left(x_{t+1}|\left(1,0\right)Av_{t},\sigma_{0,0}\right)\;\mathrm{and}\;P\left(y_{t+1}|v_{t}\right)=N\left(y_{t+1}|\left(0,1\right)Av_{t},\sigma_{1,1}\right).$$
Thus, the conditional entropy of $x_{t+1}$ and $y_{t+1}$ given $v_{t}={\left(x_{t},y_{t}\right)}^{⊤}$ are
$$H\left(x_{t+1}|x_{t},y_{t}\right)=\frac{1}{2}\log|2\pi e\sigma_{0,0}|\;\mathrm{and}\;H\left(y_{t+1}|x_{t},y_{t}\right)=\frac{1}{2}\log\left|2\pi e\sigma_{1,1}\right|.$$
With the conditional probability distribution and marginal probability distribution
$$P\left(v_{t+1}|v_{t}\right)=N\left(v_{t+1}|Av_{t},\Sigma \right)\;\mathrm{and}\;P\left(v_{t}\right)=N\left(v_{t}|A^{t}v_{0},\Sigma_{t}\right),$$
the joint probability distribution of $v_{t}$ and $v_{t+1}$ is their product
$$P\left(v_{t+1},v_{t}\right)=N\left(v_{t+1}|Av_{t},\Sigma \right)N\left(v_{t}|A^{t}v_{0},\Sigma_{t}\right).$$
Specifically, this quad-variate normal distribution is
$$\begin{array}{ccc} P(v_{t+1},v_{t}) & = & e^{-\frac{1}{2}{\left(v_{t+1}-Av_{t}\right)}^{⊤}\Sigma^{-1}\left(v_{t+1}-Av_{t}\right)-\frac{1}{2}{\left(v_{t}-A^{t}v_{0}\right)}^{⊤}\Sigma_{t}^{-1}\left(v_{t}-A^{t}v_{0}\right)}{\left(2\pi \right)}^{-2}{\left|\Sigma \right|}^{-\frac{1}{2}}{\left|\Sigma_{t}\right|}^{-\frac{1}{2}}. \end{array}$$
Applying the identities
$$v_{t+1}-Av_{t}=v_{t+1}-A^{t+1}v_{0}-A\left(v_{t}-A^{t}v_{0}\right),$$
$$\Sigma_{t}^{\prime }:=\left(\begin{array}{cc} \Sigma +A\Sigma_{t}A^{⊤} & A\Sigma_{t}\\ \Sigma_{t}A^{⊤} & \Sigma_{t} \end{array}\right)={\left(\begin{array}{cc} \Sigma^{-1} & -\Sigma^{-1}A\\ -A^{⊤}\Sigma^{-1} & \Sigma_{t}^{-1}+A\Sigma^{-1}A^{⊤} \end{array}\right)}^{-1},$$
and $\left|\Sigma |\right|\Sigma_{t}\left|=\right|\Sigma_{t}^{\prime }|$ to $P(v_{t+1},v_{t})$, we have
$$P\left(v_{t+1},v_{t}\right)=N\left(v_{t}^{\prime }|\mu_{t}^{\prime },\Sigma_{t}^{\prime }\right),$$
where
$$v_{t}^{\prime }:=\left(\begin{array}{c} v_{t+1}\\ v_{t} \end{array}\right),\;\mu_{t}^{\prime }:=\left(\begin{array}{c} A^{t+1}v_{0}\\ A^{t}v_{0} \end{array}\right),\;\Sigma_{t}^{\prime }:=\left(\begin{array}{cc} \Sigma +A\Sigma_{t}A^{⊤} & A\Sigma_{t}\\ \Sigma_{t}A^{⊤} & \Sigma_{t} \end{array}\right).$$From this joint probability distribution $P(v_{t+1},v_{t})$, we drive the marginal distributions
$$P\left(x_{t+1},x_{t}\right)=N\left(x_{t+1},x_{t}|\mu_{t,0},\Sigma_{t,0}\right),\;P\left(y_{t+1},y_{t}\right)=N\left(y_{t+1},y_{t}|\mu_{t,1},\Sigma_{t,1}\right),$$
where for i=0,1 the mean vectors and covariance matrices are defined as follows:
$$\begin{array}{ccc} \mu_{t,i} & := & \left(\begin{array}{c} e_{i}^{⊤}A^{t+1}v_{0}\\ e_{i}^{⊤}A^{t}v_{0} \end{array}\right)=\left(I_{2}⊗e_{i}\right)\mu_{t}^{\prime },\\ \Sigma_{t,i} & := & \left(\begin{array}{cc} e_{i}^{⊤}\left(\Sigma +A\Sigma_{t}A^{⊤}\right)e_{i} & e_{i}^{⊤}A\Sigma_{t}e_{i}\\ e_{i}^{⊤}\Sigma_{t}A^{⊤}e_{i} & e_{i}^{⊤}\Sigma_{t}e_{i} \end{array}\right)={\left(I_{2}⊗e_{i}\right)}^{⊤}\Sigma_{t}^{\prime }\left(I_{2}⊗e_{i}\right), \end{array}$$
with the unit vectors $e_{0}:={\left(1,0\right)}^{⊤}$, $e_{1}:={\left(0,1\right)}^{⊤}$.Thus, we have the joint entropy of $x_{t}$ and $x_{t+1}$
(8)$$\begin{array}{ccc} H(x_{t+1},x_{t}) & = & \frac{1}{2}\log\left|2\pi e\Sigma_{t,0}\right| \end{array}$$
(9)$$\begin{array}{ccc} & = & \frac{1}{2}\log{\left(2\pi e\right)}^{2}\left|e_{0}^{⊤}\left(\Sigma +A\Sigma_{t}A^{⊤}\right)e_{0}e_{0}^{⊤}\Sigma_{t}e_{0}-{\left(e_{0}^{⊤}A\Sigma_{t}e_{0}\right)}^{2}\right|, \end{array}$$
and the marginal distribution of $x_{t}$
(10)$$H\left(x_{t}\right)=\frac{1}{2}\log2\pi e\left|e_{0}^{⊤}\Sigma_{t}e_{0}\right|.$$
Using these, we have the conditional entropy
(11)$$H\left(x_{t+1}|x_{t}\right)=\frac{1}{2}\log2\pi \frac{e_{0}^{⊤}\left(\Sigma +A\Sigma_{t}A^{⊤}\right)e_{0}e_{0}^{⊤}\Sigma_{t}e_{0}-{\left(e_{0}^{⊤}A\Sigma_{t}e_{0}\right)}^{2}}{|e_{0}^{⊤}\Sigma_{t}e_{0}|}.$$
By the stability of VAR, the Lyapunov Equation (Equation 4) holds, and this conditional entropy in the limit t→∞ is
(12)$$\begin{array}{ccc} \lim_{t\to \infty }H\left(x_{t+1}|x_{t}\right) & = & \frac{1}{2}\log2\pi \frac{{\left(e_{0}^{⊤}\hat{\Sigma }e_{0}\right)}^{2}-{\left(e_{0}^{⊤}A\hat{\Sigma }e_{0}\right)}^{2}}{|e_{0}^{⊤}\hat{\Sigma }e_{0}|}. \end{array}$$
Applying Definition 2 and denoting by entries in the stationary covariance matrix $\hat{\Sigma }=\left(\begin{array}{cc} {\hat{\sigma }}_{0,0} & {\hat{\sigma }}_{0,1}\\ {\hat{\sigma }}_{1,0} & {\hat{\sigma }}_{1,1} \end{array}\right)$, we have
(13)$$G_{0}=2T_{y\to x}=\log\frac{{\left({\hat{\sigma }}_{0,0}\right)}^{2}-{\left(a_{0,0}{\hat{\sigma }}_{0,0}+a_{0,1}{\hat{\sigma }}_{1,0}\right)}^{2}}{{\hat{\sigma }}_{0,0}\sigma_{0,0}}.$$
Similarly, we have
(14)$$G_{1}=2T_{x\to y}=\log\frac{{\left({\hat{\sigma }}_{1,1}\right)}^{2}-{\left(a_{1,0}{\hat{\sigma }}_{0,1}+a_{1,1}{\hat{\sigma }}_{1,1}\right)}^{2}}{{\hat{\sigma }}_{1,1}\sigma_{1,1}}.$$
Let us define for i=0,1
(15)$$\delta_{i}:={\left({\hat{\sigma }}_{i,i}\right)}^{2}-{\left(a_{i,i}{\hat{\sigma }}_{i,i}+a_{i,1-i}{\hat{\sigma }}_{1-i,i}\right)}^{2}\;\mathrm{and}\;\delta_{i}^{\prime }:={\hat{\sigma }}_{i,i}\sigma_{i,i}.$$
By the Lyapunov equation, $\sigma_{i,i}={\hat{\sigma }}_{i,i}-e_{i}^{⊤}A\hat{\Sigma }A^{⊤}e_{i}$. Applying this to $\delta_{i}^{\prime }$, we have
(16)$$\delta_{i}={\hat{\sigma }}_{i,i}^{2}-e_{i}^{⊤}A\left(\begin{array}{cc} {\hat{\sigma }}_{i,i}^{2} & {\hat{\sigma }}_{i,i}{\hat{\sigma }}_{1-i,i}\\ {\hat{\sigma }}_{i,i}{\hat{\sigma }}_{1-i,i} & {\hat{\sigma }}_{1-i,i}^{2} \end{array}\right)A^{⊤}e_{i},$$
(17)$$\delta_{i}^{\prime }={\hat{\sigma }}_{i,i}^{2}-e_{i}^{⊤}A\left(\begin{array}{cc} {\hat{\sigma }}_{i,i}{\hat{\sigma }}_{0,0} & {\hat{\sigma }}_{i,i}{\hat{\sigma }}_{0,1}\\ {\hat{\sigma }}_{i,i}{\hat{\sigma }}_{1,0} & {\hat{\sigma }}_{i,i}{\hat{\sigma }}_{1,1} \end{array}\right)A^{⊤}e_{i}.$$
As $\delta_{i}-\delta_{i}^{\prime }=a_{i,1-i}^{2}\det\left(\hat{\Sigma }\right)$ and $G_{0}=\log\left(1+\frac{\delta_{i}-\delta_{i}^{\prime }}{\delta_{i}^{\prime }}\right)$, we have
$$\begin{array}{c} G_{i}=\log\left(1+\frac{a_{i,1-i}^{2}\det\left(\hat{\Sigma }\right)}{{\hat{\sigma }}_{i,i}\sigma_{i,i}}\right). \end{array}$$ □

The Granger causality has its lower and upper bounds in theory. Although these bounds may be further narrowed by considering the stability of the VAR, what follows below are the theoretical bounds regardless of the stability of the VAR.

**Lemma** **3**(The upper and lower bound for Granger causality). *For each i=0,1, Granger causality $G_{i}$ has the following bounds:*
(18)$$0\leq G_{i}\leq \log\gamma_{i},$$
*where $\gamma_{i}:=\frac{{\hat{\sigma }}_{i,i}}{\sigma_{i,i}}\geq 1$ due to the Lyapunov Equation (Equation 4). The lower bound $G_{i}=0$ is given only if*
(19)$$a_{i,1-i}^{2}\det\left(\hat{\Sigma }\right)=0.$$
*The upper bound $G_{i}=\log\gamma_{i}$ is given only if*
(20)$$a_{i,i}{\hat{\sigma }}_{i,i}+a_{i,1-i}{\hat{\sigma }}_{1-i,i}=0.$$

**Proof.** As the stationary covariance matrix is (semi-)positive definite, $\det\left(\hat{\Sigma }\right)\geq 0$. Thus, the lower bound of Granger causality is $G_{i}\geq \log\left(1\right)=0$ and this bound is only reacheable when $a_{i,1-i}^{2}\det\left(\hat{\Sigma }\right)=0$.Modifying (Equation 13) and (Equation 14), for i=0,1 we have
(21)$${\left(a_{i,i}{\hat{\sigma }}_{i,i}+a_{i,1-i}{\hat{\sigma }}_{1-i,i}\right)}^{2}={\hat{\sigma }}_{i,i}\left({\hat{\sigma }}_{i,i}-\sigma_{i,i}e^{G_{i}}\right).$$
As ${\left(a_{i,i}{\hat{\sigma }}_{i,i}+a_{i,1-i}{\hat{\sigma }}_{1-i,i}\right)}^{2}\geq 0$ and ${\hat{\sigma }}_{i,i}>0$,
$$G_{i}\leq \log\gamma_{i}.$$
This upper bound holds only if $a_{i,i}{\hat{\sigma }}_{i,i}+a_{i,1-i}{\hat{\sigma }}_{1-i,i}=0$. □

The upper bound Lemma 3 can also be obtained by the following information-theoretic identity:$$\lim_{t\to \infty }I\left(x_{t-1};x_{t}\right)+I\left(x_{t};x_{t-1}|x_{t-1}\right)=\lim_{t\to \infty }I\left(x_{t};x_{t-1},x_{t-1}\right),$$
where $\lim_{t\to \infty }I\left(x_{t};x_{t-1}|x_{t-1}\right)=\frac{1}{2}G_{0}$ is the transfer entropy, $\lim_{t\to \infty }I\left(x_{t};x_{t-1},x_{t-1}\right)=\frac{1}{2}\log\frac{{\hat{\sigma }}_{0,0}}{\sigma_{0,0}}$, and
$$\lim_{t\to \infty }I\left(x_{t-1};x_{t}\right)=\frac{1}{2}\log\frac{{\hat{\sigma }}_{0,0}^{2}}{\left|\begin{array}{cc} \hat{\Sigma } & \hat{\Sigma }A^{⊤}\\ A\hat{\Sigma } & \hat{\Sigma } \end{array}\right|}=\frac{1}{2}\left(\log\gamma_{0}-G_{0}\right).$$

## 4. Stability and Constraints of VAR

In this study, we primarily consider the class of stable VAR models with a proper set of parameters. In this class, the statistical nature of any VAR is characterized by the base covariance matrix $\Sigma \in R^{2\times 2}$, coefficient matrix $A\in R^{2\times 2}$, and stationary covariance matrix $\hat{\Sigma }\in R^{2\times 2}$. Let us denote the set of (strictly) positive definite matrices by
$$R_{+}^{2\times 2}:=\left\{M\in R^{2\times 2}|\det\left(M\right)>0\;\mathrm{and}\;\mathrm{tr}\left(M\right)>0\right\},$$
and the set of coefficient matrices of stable VAR models
$$R_{*}^{2\times 2}:=\left\{M\in R^{2\times 2}|-1+|\mathrm{tr}\left(M\right)|<\det\left(M\right)<1\right\}.$$
We will briefly show that the stable set $R_{*}^{2\times 2}$ includes all and only coefficient matrices of stable bivariate VAR models.

With this notation of the set of matrices, the two conditions that any proper VAR model needs to satisfy are as follows.

**Stability** Any stable VAR model has both of the eigenvalues $\lambda_{0},\lambda_{1}$ of its coefficient matrix *A* meeting $|\lambda_{0}\left|,\right|\lambda_{1}|<1$.**Properness** To have a proper (non-degenerated) bivariate normal distribution in a VAR model, its base covariance matrix Σ and stationary covariance matrix $\hat{\Sigma }$ need to satisfy $\Sigma ,\hat{\Sigma }\in R_{+}^{2\times 2}$. The set of positive-definite matrices is equivalently written with the entries of the following matrix:
(22)$$R_{+}^{2\times 2}=\left\{C\in R^{2\times 2}|C_{0,0}>0,C_{1,1}>0,\;\mathrm{and}\;C_{0,0}C_{1,1}-C_{0,1}C_{1,0}>0\right\}.$$

### 4.1. Stability of VAR

As stated previously in Section 2.2, the stability of VAR is primarily characterized by the eigenvalues of the coefficient matrix *A*. However, this condition is equivalent to $A\in R_{*}^{2\times 2}$, as shown by the following lemma.

**Lemma** **4.**
*A given bivariate VAR model with its coefficient matrix $A\in R^{2\times 2}$ is stable if and only if*
(23)$$|\mathrm{tr}\left(A\right)|-1<\det\left(A\right)<1.$$


**Proof.** Let λ be an eigenvalue of the coefficient matrix *A*. Such an eigenvalue then satisfies
(24)$$f\left(\lambda \right)=\left|A-\lambda I_{2}\right|=\lambda^{2}-\mathrm{tr}\left(A\right)\lambda +\det\left(A\right)=0.$$
If a VAR is stable, this eigenvalue needs to satisfy |λ|<1. As (Equation 24) is rewritten by
(25)$$f\left(\lambda \right)={\left(\lambda -\frac{1}{2}\mathrm{tr}\left(A\right)\right)}^{2}-\frac{1}{4}\left(\mathrm{tr}{\left(A\right)}^{2}-4\det\left(A\right)\right),$$
we analyze this condition on (Equation 24) for the following two cases with λ being real or non-real:
If λ is real, this stability condition is equivalent to
(26)$$\mathrm{tr}{\left(A\right)}^{2}\leq 4\det\left(A\right),f\left(1\right)>0,\;f\left(-1\right)>0,\;\left|\mathrm{tr}\left(A\right)\right|<2.$$If λ is not real, this stability condition is equivalent with
(27)$$\mathrm{tr}{\left(A\right)}^{2}<4\det\left(A\right),\;{\left|\lambda \right|}^{2}<1.$$If λ of (Equation 24) is non-real (Case 2), λ (and its conjugate) is
(28)$$\lambda =\frac{1}{2}\mathrm{tr}\left(A\right)\pm \frac{j}{2}\sqrt{|\mathrm{tr}{\left(A\right)}^{2}-4\det\left(A\right)|},$$
with the imaginary unit denoted by *j*.With the inequality (Equation 27), the stability condition in this case is
(29)$${\left(\frac{\mathrm{tr}\left(A\right)}{2}\right)}^{2}<{\left|\lambda \right|}^{2}=\det\left(A\right)<1.$$If λ of (Equation 24) is real, $\mathrm{tr}{\left(A\right)}^{2}-4\det\left(A\right)\geq 0$ and
(30)$$\begin{array}{c} f\left(1\right)=1-\mathrm{tr}\left(A\right)+\det\left(A\right)>0 \end{array}$$
(31)$$\begin{array}{c} f\left(-1\right)=1+\mathrm{tr}\left(A\right)+\det\left(A\right)>0 \end{array}$$
(32)$$\begin{array}{c} |\mathrm{tr}\left(A\right)|<2. \end{array}$$
Combining (30) and (31), we have $|\mathrm{tr}\left(A\right)|-1<\det\left(A\right)$. This inequality with (Equation 26),
(33)$$C_{0}<\det\left(A\right)\leq {\left(\frac{\mathrm{tr}\left(A\right)}{2}\right)}^{2}<C_{1},$$
where $C_{0}:=\left|\mathrm{tr}\left(A\right)\right|-1$ and $C_{1}:=\min\left(1,{\left(\frac{1+\det\left(A\right)}{2}\right)}^{2}\right)$. Find for an arbitrary $A\in R^{2\times 2}$ we have the following two inequalities:
(34)$$|\mathrm{tr}\left(A\right)|-1\leq {\left(\frac{1}{2}\mathrm{tr}\left(A\right)\right)}^{2},$$
and
(35)$$\det\left(A\right)\leq {\left(\frac{1+\det\left(A\right)}{2}\right)}^{2}.$$
The inequality (Equation 34) holds equality for and only for $\mathrm{tr}\left(A\right)=2$, and the inequality (Equation 34) holds equality for and only for $\det\left(A\right)=1$. As both of these equality conditions do not hold under (Equation 29), (Equation 29) is equivalent to
(36)$$C_{0}<{\left(\frac{\mathrm{tr}\left(A\right)}{2}\right)}^{2}<\det\left(A\right)<C_{1}.$$
Integrating the two inequalities (Equation 33) for real λ and (Equation 36) for non-real λ, the VAR with the coefficient matrix *A* is stable if
(37)$$C_{0}<\det\left(A\right)<C_{1}$$
and
(38)$$C_{0}<{\left(\frac{\mathrm{tr}\left(A\right)}{2}\right)}^{2}<C_{1}.$$
As the inequality (Equation 37) implies $0<\frac{1+\det\left(A\right)}{2}<1$ and $\mathrm{tr}\left(A\right)<2$, (Equation 38) is equivalent to
(39)$${\left(\frac{\mathrm{tr}\left(A\right)}{2}\right)}^{2}<{\left(\frac{1+\det\left(A\right)}{2}\right)}^{2}.$$
As the upper bound for $\det\left(A\right)$ in (Equation 37) can be implied by $\det\left(A\right)<1$, it is equivalent to
(40)$$|\mathrm{tr}\left(A\right)|-1<\det\left(A\right)<1.$$
Thus, the pair of inequalities (Equation 37) and (Equation 38) for *A* is equivalent to the single inequality (Equation 40) for *A*. □

### 4.2. Stability and Existence of the Solution for the Lyapunov Equation

Intuitively, it would be reasonable if there was a stationary covariance matrix $\hat{\Sigma }\in R_{+}^{2\times 2}$ satisfying the Lyapunov Equation (Equation 4), if the coefficient matrix is $A\in R_{*}^{2\times 2}$. However, this is not trivial, as the opposite may not be always true: the existence of $\hat{\Sigma }\in R_{+}^{2\times 2}$ does not imply $A\in R_{*}^{2\times 2}$. This relationship between *A* and $\hat{\Sigma }$ is stated by the following Theorem 1.

**Theorem** **1.**
*There is a stationary covariance matrix $\hat{\Sigma }\in R_{+}^{2\times 2}$ satisfying the Lyapunov Equation (Equation 4), if the coefficient matrix is $A\in R_{*}^{2\times 2}$. However, the existence of $\hat{\Sigma }\in R_{+}^{2\times 2}$ does not imply $A\in R_{*}^{2\times 2}$.*


**Proof.** Find the identity
(41)$$\begin{array}{cc} \det\left(I_{4}-A⊗A\right) & =\det\left(\left(I_{2}-a_{0,0}A\right)\left(I_{2}-a_{1,1}A\right)-a_{0,1}a_{1,0}A^{2}\right)\\ & =\det\left(I_{2}-A\mathrm{tr}(A)+A^{2}\det(A)\right)\\ & ={\left(1-\det(A)\right)}^{2}\left(\left(1-a_{0,1}-a_{0,0}^{2}\right)\left(1-a_{0,1}-a_{1,1}^{2}\right)-a_{0,1}a_{1,0}\mathrm{tr}(A)\right)\\ & ={\left(1-\det(A)\right)}^{2}\left({\left(1+\det(A)\right)}^{2}-\mathrm{tr}{\left(A\right)}^{2}\right). \end{array}$$
By Lemma 4 and (41), we have $\det\left(I_{4}-A⊗A\right)>0$. Thus, Lyapunov Equation (Equation 5) has the solution for $\hat{\Sigma }$, as the matrix $\left(I_{4}-A⊗A\right)$ is invertible. The converse of this theorem does not hold, as we construct a counter-example of the coefficient matrix *A* such that $\det\left(I_{4}-A⊗A\right)<0$, with which there is a $\hat{\Sigma }\in R_{+}^{2\times 2}$, but such a VAR is not stable. □

## 5. Design of Bivariate Timeseries Given GCs

The goal of this study was to derive a design principle of bivariate timeseries generated by a VAR model with the desired correlation and two types of Granger causality. In this section, we explore the inter-dependent relationships among the variables in the VAR. This analysis revealed a trade-off limitation in designing these variables of timeseries. Specifically, a timeseries with a certain range of desired Granger causality cannot be realized by a stable VAR, in which no stationary covariance is defined in theory.

The set of parameters in any stable VAR model includes

The coefficient matrix *A*;The base covariance matrix Σ;The stationary covariance matrix $\hat{\Sigma }$; andThe two types of Granger causality $G_{0},G_{1}$.

There are equality constraints on these variables:The variables $A,\Sigma ,\hat{\Sigma }$ need to satisfy the Lyapunov Equation (Equation 4).Granger causality $G_{i}$ (i=0,1) is the function of $a_{i,1-i}$, $\sigma_{i,i}$, and $\hat{\Sigma }$ (Lemma 2).

Besides, it is important to know the feasibility of a set of parameters in VAR, which constrains the range of these variables:Stability: $A\in R_{*}^{2\times 2}$ (Section 4);Properness: $\Sigma ,\hat{\Sigma }\in R_{+}^{2\times 2}$ (Section 4) and $\sigma_{i,i}\leq {\hat{\sigma }}_{i,i}$ due to the existence of a solution for the Lyapunov equation; andThe bound for each Granger causality: $G_{i}\in \left[0,\log\gamma_{i}\right]$ (Lemma 3).

The Lyapunov Equation (Equation 4) on the matrices can be decomposed into the three equations on the scalar variables as follows. For a coefficient matrix $A=\left(\begin{array}{cc} a_{0,0} & a_{0,1}\\ a_{1,0} & a_{1,1} \end{array}\right)$, let us define two vectors by
$$a_{0}:=\left(\begin{array}{c} a_{0,0}\\ a_{0,1} \end{array}\right),\;a_{1}:=\left(\begin{array}{c} a_{1,0}\\ a_{1,1} \end{array}\right).$$
The Lyapunov equation is then equivalently written with these vectors $a_{0},a_{1}$ by the set of the three equations
(42)$$\begin{array}{ccc} {\hat{\sigma }}_{0,0}-\sigma_{0,0} & = & a_{0}^{⊤}\hat{\Sigma }a_{0} \end{array}$$
(43)$$\begin{array}{ccc} {\hat{\sigma }}_{1,1}-\sigma_{1,1} & = & a_{1}^{⊤}\hat{\Sigma }a_{1} \end{array}$$
(44)$$\begin{array}{ccc} {\hat{\sigma }}_{0,1}-\sigma_{0,1} & = & a_{0}^{⊤}\hat{\Sigma }a_{1}. \end{array}$$
Equations (Equation 42) and (43) above imply that each of the vectors $a_{0}$ and $a_{1}$ are on an ellipsis on each of their planes. This gives the lower bound for ${\hat{\sigma }}_{i,i}\geq \sigma_{i,i}$ (i.e., one condition of the properness above), as $x^{⊤}\hat{\Sigma }x\geq 0$ for any $x\in R^{2}$ with a positive-definite matrix $\hat{\Sigma }$.

Fixing $G_{0}$ and $G_{1}$ imposes each of the two vectors $a_{0}$ and $a_{1}$ on the two parallel lines by
(45)$${\left(a_{i}^{⊤}{\hat{\sigma }}_{i}\right)}^{2}=\tau_{i}^{2},$$
where
$${\hat{\sigma }}_{i}:={\left({\hat{\sigma }}_{i,0},{\hat{\sigma }}_{i,1}\right)}^{⊤},\;\tau_{i}^{2}:={\hat{\sigma }}_{i,i}^{2}\left(1-\gamma_{i}^{-1}e^{G_{i}}\right).$$
Thus, the solution of $a_{i}$ which satisfies the Lyapunov equation and the fixed Granger causality is the four intersections of the ellipsis and the two parallel lines (Figure 2). This ellipsis is obtained by scaling and shearing transformation to the standard circle $a_{0,0}^{2}+a_{0,1}^{2}=1$. This observation gives the angular parametrization of the solution vector ${\left(a_{0,0},a_{0,1}\right)}^{⊤}$, which is explicitly stated by Lemma 5 in the next section.

### 5.1. Solution A of the Lyapunov Equality Given $\hat{\Sigma }$, $G_{0}$, and $G_{1}$

In what follows, we start with the derivation of the coefficient matrix *A* as a root of the equality constraint by the Lyapunov Equation (Equation 4) and the Granger causality, for a fixed proper $\hat{\Sigma }$, $\sigma_{i,i}$ and $G_{i}$ for each i=0,1. The following Lemma 5 gives a necessary condition for the coefficient matrix $A\in R^{2\times 2}$ to satisfy the equality conditions above. Note, however, that such a solution *A* in this equation does not guarantee the stability of the corresponding VAR (i.e., $A\in R_{*}^{2\times 2}$). This sufficiency is explored in Section 5.2.

**Lemma** **5.**
*For a given set of parameters, a positive-definite matrix $\hat{\Sigma }=\left(\begin{array}{cc} {\hat{\sigma }}_{0,0} & {\hat{\sigma }}_{0,1}\\ {\hat{\sigma }}_{1,0} & {\hat{\sigma }}_{1,1} \end{array}\right)\in R_{+}^{2\times 2}$, $\sigma_{i,i}\in \left(0,{\hat{\sigma }}_{i,i}\right),G_{i}\in \left[0,\log\gamma_{i}\right]$ for each i=0,1, suppose that a coefficient matrix $A=\left(\begin{array}{cc} a_{0,0} & a_{0,1}\\ a_{1,0} & a_{1,1} \end{array}\right)\in R^{2\times 2}$, satisfies the set of the equations*
(46)$$\left\{\begin{array}{c} a_{0}^{⊤}\hat{\Sigma }a_{0}={\hat{\sigma }}_{0,0}-\sigma_{0,0}\\ a_{1}^{⊤}\hat{\Sigma }a_{1}={\hat{\sigma }}_{1,1}-\sigma_{1,1}\\ {\left({\hat{\sigma }}_{0}^{⊤}a_{0}\right)}^{2}=\tau_{0}^{2}\\ {\left({\hat{\sigma }}_{1}^{⊤}a_{1}\right)}^{2}=\tau_{1}^{2} \end{array}\right.,$$
*where for i=0,1*
$${\hat{\sigma }}_{i}:={\left({\hat{\sigma }}_{i,0},{\hat{\sigma }}_{i,1}\right)}^{⊤},\;\tau_{i}^{2}:={\hat{\sigma }}_{i,i}^{2}\left(1-\gamma_{i}^{-1}e^{G_{i}}\right).$$
*Any coefficient matrix A of a root of this Equation (Equation 46) is in the form*
(47)$$A={\left(\begin{array}{cc} S_{0}\left(\begin{array}{c} \cos\theta_{0}\\ \sin\theta_{0} \end{array}\right)\; & P_{2}S_{1}\left(\begin{array}{c} \cos\theta_{1}\\ \sin\theta_{1} \end{array}\right) \end{array}\right)}^{⊤},$$
*where each pair of the angles $\theta_{0}\in \left[0,2\pi \right)$ and $\theta_{1}\in \left[0,2\pi \right)$ takes one of the two or four pairs satisfying for each i=0,1*
(48)$$\sin^{2}\theta_{i}=\frac{e^{G_{i}}-1}{\gamma_{i}-1}$$
*and*
$$P_{2}:=\left(\begin{array}{cc} 0 & 1\\ 1 & 0 \end{array}\right),\;S_{i}:=\sqrt{1-\gamma_{i}^{-1}}\left(\begin{array}{cc} 1 & -\frac{{\hat{\sigma }}_{i,1-i}}{{\hat{\sigma }}_{i,i}}\\ 0 & 1 \end{array}\right)\left(\begin{array}{cc} 1 & 0\\ 0 & \frac{{\hat{\sigma }}_{i,i}}{\sqrt{\det\left(\hat{\Sigma }\right)}} \end{array}\right).$$


**Proof.** Find that the following pair of equations in (Equation 46) is symmetric under exchange of i=0,1:
(49)$$\left\{\begin{array}{c} a_{i}^{⊤}\hat{\Sigma }a_{i}={\hat{\sigma }}_{i,i}-\sigma_{i,i}\\ {\left({\hat{\sigma }}_{i}^{⊤}a_{i}\right)}^{2}=\tau_{i}^{2} \end{array}\right..$$
Thus, we solve this for i=0 below, and it holds for i=1.Solving the second equation of (Equation 49) for $a_{0,0}$, we have
(50)$$a_{0,0}=\frac{\pm \tau_{0}-a_{0,1}{\hat{\sigma }}_{0,1}}{{\hat{\sigma }}_{0,0}}.$$
Inserting this into the first equation of (Equation 49), we have
(51)$$a_{0,1}^{2}=\frac{{\hat{\sigma }}_{0,0}^{2}\left(1-\gamma_{0}^{-1}\right)-\tau_{0}^{2}}{\det\left(\hat{\Sigma }\right)}.$$Inserting $\tau_{0}^{2}={\hat{\sigma }}_{0,0}^{2}\left(1-\gamma_{0}^{-1}e^{G_{0}}\right)$, we have
(52)$$a_{0,1}=\pm \sqrt{\frac{{\hat{\sigma }}_{0,0}\sigma_{0,0}\left(e^{G_{0}}-1\right)}{\det\left(\hat{\Sigma }\right)}}.$$
Inserting this into (Equation 50), we have at most four vectors $a_{0}={\left(a_{0,0},a_{0,1}\right)}^{⊤}$ as the solution of (Equation 49) for i=1:
(53)$$a_{0}=\left(\begin{array}{c} c_{0}-\frac{{\hat{\sigma }}_{0,1}}{{\hat{\sigma }}_{0,0}}d_{0}\\ d_{0} \end{array}\right),\;\left(\begin{array}{c} c_{0}+\frac{{\hat{\sigma }}_{0,1}}{{\hat{\sigma }}_{0,0}}d_{0}\\ -d_{0} \end{array}\right),\;\left(\begin{array}{c} -c_{0}-\frac{{\hat{\sigma }}_{0,1}}{{\hat{\sigma }}_{0,0}}d_{0}\\ d_{0} \end{array}\right),\;\left(\begin{array}{c} -c_{0}+\frac{{\hat{\sigma }}_{0,1}}{{\hat{\sigma }}_{0,0}}d_{0}\\ -d_{0} \end{array}\right),$$
where for i=0,1
$$c_{i}:=\sqrt{1-\gamma_{i}^{-1}e^{G_{i}}},\;d_{i}:=\sqrt{\frac{{\hat{\sigma }}_{i,i}\sigma_{i,i}\left(e^{G_{i}}-1\right)}{\det\left(\hat{\Sigma }\right)}}.$$
By symmetry to i=0,1, there are at most four vectors as the solution of (Equation 49) for i=1:
(54)$$P_{2}a_{1}=\left(\begin{array}{c} a_{1,1}\\ a_{1,0} \end{array}\right)=\left(\begin{array}{c} c_{1}-\frac{{\hat{\sigma }}_{1,0}}{{\hat{\sigma }}_{1,1}}d_{1}\\ d_{1} \end{array}\right),\;\left(\begin{array}{c} c_{1}+\frac{{\hat{\sigma }}_{1,0}}{{\hat{\sigma }}_{1,1}}d_{1}\\ -d_{1} \end{array}\right),\;\left(\begin{array}{c} -c_{1}-\frac{{\hat{\sigma }}_{1,0}}{{\hat{\sigma }}_{1,1}}d_{1}\\ d_{1} \end{array}\right),\;\left(\begin{array}{c} -c_{1}+\frac{{\hat{\sigma }}_{1,0}}{{\hat{\sigma }}_{1,1}}d_{1}\\ -d_{1}. \end{array}\right)$$Find that these four solution vectors parameterized by
$$a_{0}=S_{0}\left(\begin{array}{c} \cos\theta_{0}\\ \sin\theta_{0} \end{array}\right)\;\mathrm{and}\;a_{1}=P_{2}S_{1}\left(\begin{array}{c} \cos\theta_{1}\\ \sin\theta_{1} \end{array}\right),$$
satisfy (Equation 49), if $\left(\theta_{0},\theta_{1}\right)$ holds (Equation 48), with the trigonometric identity $\cos^{2}\theta_{i}+\sin^{2}\theta_{i}=1$ and
(55)$$S_{0}^{⊤}\hat{\Sigma }S_{0}=\left({\hat{\sigma }}_{0,0}-\sigma_{0,0}\right)I_{2}\;\mathrm{and}\;S_{1}^{⊤}P_{2}^{⊤}\hat{\Sigma }P_{2}S_{1}=\left({\hat{\sigma }}_{1,1}-\sigma_{1,1}\right)I_{2}.$$□

### 5.2. Sufficiency of the Solution

For a given set of parameters, Lemma 5 in the previous section gives a set of solutions of the coefficient matrix *A* for the Lyapunov equation. Note that not all of these solutions *A* are feasible, in the sense that they satisfy all constraints such as the stability of $A\in R_{*}^{2\times 2}$ and the properness of $\Sigma \in R_{+}^{2\times 2}$, in which Σ can be derived from Lyapunov Equation (Equation 4) given *A* and $\hat{\Sigma }$. The following lemmas provide the sufficient condition for a solution *A* by checking the properness of Σ and the stability of *A*.

**Lemma** **6.**
*Suppose A is a solution of Equation (Equation 46) in Lemma 5, represented by a pair of $\left(\theta_{0},\theta_{1}\right)$. In this case, $\Sigma \in R_{+}^{2\times 2}$, if and only if*
(56)$$\cos\hat{\eta }-{\left(\gamma_{0}\gamma_{1}\right)}^{-\frac{1}{2}}\leq {\hat{\gamma }}_{0}{\hat{\gamma }}_{1}\cos\left(\hat{\eta }-\theta_{0}-\theta_{1}\right)\leq \cos\hat{\eta }+{\left(\gamma_{0}\gamma_{1}\right)}^{-\frac{1}{2}},$$
*where $\hat{\eta }\in \left[0,2\pi \right]$ is the angler parametrization of the correlation coefficient defined by $\cos\hat{\eta }:=\frac{{\hat{\sigma }}_{0,1}}{\sqrt{{\hat{\sigma }}_{0,0}{\hat{\sigma }}_{1,1}}}$ and ${\hat{\gamma }}_{i}=\sqrt{1-\gamma_{i}^{-1}}$.*


**Proof.** Applying the polar representation of the $a_{0},a_{1}$ in (Equation 47) in Lemma 5 to the third Equation (44) of the Lyapunov equation, we have
(57)$${\hat{\sigma }}_{0,1}-\sigma_{0,1}=\sqrt{\left({\hat{\sigma }}_{0,0}-\sigma_{0,0}\right)\left({\hat{\sigma }}_{1,1}-\sigma_{1,1}\right)}\cos\left(\hat{\eta }-\theta_{0}-\theta_{1}\right).$$
By the positive definiteness of Σ, $\sigma_{0,1}^{2}\leq \sigma_{0,0}\sigma_{1,1}$. This inequality applied to (Equation 57) gives the lemma. □

If we have
$${\hat{\gamma }}_{0}{\hat{\gamma }}_{1}\leq \cos\hat{\eta }+{\left(\gamma_{0}\gamma_{1}\right)}^{-\frac{1}{2}}\;\;\mathrm{and}\;\;\cos\hat{\eta }-{\left(\gamma_{0}\gamma_{1}\right)}^{-\frac{1}{2}}\leq -{\hat{\gamma }}_{0}{\hat{\gamma }}_{1},$$
Equation (Equation 56) holds for any pair of angles $\left(\theta_{0},\theta_{1}\right)$. This condition is
$$|\cos\hat{\eta }|\leq {\left(\gamma_{0}\gamma_{1}\right)}^{-\frac{1}{2}}-{\hat{\gamma }}_{0}{\hat{\gamma }}_{1},$$
or
(58)$$|{\hat{\sigma }}_{0,1}|\leq \sqrt{\sigma_{0,0}\sigma_{1,1}}-\sqrt{\left({\hat{\sigma }}_{0,0}-\sigma_{0,0}\right)\left({\hat{\sigma }}_{1,1}-\sigma_{1,1}\right)}.$$
On the other hand, (Equation 56) holds for any $0<\hat{\eta }<\pi$ (equivalently $-1<\cos\hat{\eta }<1$) if
(59)$$\frac{\sqrt{\gamma_{0}\gamma_{1}}-1}{\sqrt{\left(\gamma_{0}-1\right)\left(\gamma_{1}-1\right)}}\leq \cos\left(\theta_{0}+\theta_{1}\right)\leq \frac{\sqrt{\gamma_{0}\gamma_{1}}+1}{\sqrt{\left(\gamma_{0}-1\right)\left(\gamma_{1}-1\right)}}.$$
These bounds (Equation 58) and (Equation 59) mean that the range of feasible GCs ($\theta_{0},\theta_{1}$) and the range of feasible correlation $\cos\hat{\eta }$ are in a trade-off relationship in general.

**Lemma** **7.**
*The VAR with the correlation $|\cos(\hat{\eta })|<1$ and Granger causality $\theta_{0},\theta_{1}$ in the angular form is stable if and only if*
(60)$$-\sin\hat{\eta }+\left|{\hat{\gamma }}_{0}\sin\left(\hat{\eta }-\theta_{0}\right)+{\hat{\gamma }}_{1}\sin\left(\hat{\eta }-\theta_{1}\right)\right|<{\hat{\gamma }}_{0}{\hat{\gamma }}_{1}\sin\left(\hat{\eta }-\theta_{0}-\theta_{1}\right)<\sin\hat{\eta }.$$


**Proof.** Using the angular notation of the solution *A* with $\theta_{0},\theta_{1}$, we have
$$\det\left(A\right)={\hat{\gamma }}_{0}{\hat{\gamma }}_{1}\frac{\sin\left(\hat{\eta }-\theta_{0}-\theta_{1}\right)}{\sin\hat{\eta }}\;\mathrm{and}\;\mathrm{tr}\left(A\right)=\frac{{\hat{\gamma }}_{0}\sin\left(\hat{\eta }-\theta_{0}\right)+{\hat{\gamma }}_{1}\sin\left(\hat{\eta }-\theta_{1}\right)}{\sin\hat{\eta }}.$$
Inserting these into stability condition (Equation 23), we have the inequality (Equation 60). □

As well as Lemma 6, Lemma 7 leads the trade-off relationship between $\hat{\Sigma }$ in the angle from $\hat{\eta }$ and *A* in the angle from $\theta_{0},\theta_{1}$. In general, this bound further narrows the upper and lower bounds given by Lemma 3. In general, correlation is limited to close to zero if one wishes for higher Granger causality. On the other hand, the two types of Granger causality are limited to close to zero if one wishes for a higher correlation in the absolute value.

## 6. Concluding Remarks

### 6.1. Summary and Potential Usage of the Algorithm

In this paper, we explored the relationship between the VAR parameters and timeseries statistics (Figure 1), and identified the trade-off limitation between the stationary covariance $\hat{\Sigma }$ and Granger causality $G_{0},G_{1}$ (Lemma 6). This suggests that the following Algorithm 1 will generate a timeseries with desired statistics.
**Algorithm 1:** Compute a VAR parameter set for the desired statistics **Data**: Desired timeseries statistics $\left(G_{0},G_{1},\hat{\Sigma },\sigma_{0,0},\sigma_{1,1}\right)$ in the feasible range satisfy both inequalities (Equation 56) in Lemma 6 and (Equation 60) in Lemma 7.
**_1_** Derive four sets of VAR parameters (A,Σ) for the given timeseries statistics $\left(G_{0},G_{1},\hat{\Sigma },\sigma_{0,0},\sigma_{1,1}\right)$ by Lemma 5
**_2_** Choose one of the four sets of VAR parameters (A,Σ).
 **Result**: The VAR parameters (A,Σ).


This data-generation algorithm can be used to generate surrogate data [19], which can be used to test whether an empirical timeseries is a sample from a VAR with a given correlation and Granger causality. This algorithm is also useful in analyzing to what extent a class of VAR timeseries varies under the same statistics.

### 6.2. Validity of Granger Causality Estimated on Empirical Timeseries

Our analysis also warns that not all Granger causality (or transfer entropy) is “valid”, in the sense that its underlying VAR model is not stable and thus the Granger causality is undefined in theory. In theory, we can identify some value of Granger causality for a finite empirical time series, which is generated by an underlying unstable VAR model without any stationary statistics. Such timeseries statistics will diverge in the long run, but it may be difficult to identify this with a finite empirical timeseries. This asymmetry—namely, that the Granger causality can be calculated numerically but does not guarantee the stability of the underlying VAR—is explicitly demonstrated by Theorem 1. For a given empirical timeseries $v=\left(v_{0},v_{1},\ldots ,v_{T}\right)\in R^{2\times (T+1)}$, one should calculate not just the Granger causality but also its validity by checking (1) $A\in R_{*}^{2\times 2}$, (2) $\Sigma \in R_{+}^{2\times 2}$, and (3) $\Sigma_{0,1}\leq {\hat{\Sigma }}_{0,1}$, by calculating the maximum likelihood estimator of (A(v),Σ(v)), such as
$$A\left(v\right):=V_{1,0}V_{0,0}^{-1}\;\mathrm{and}\;\Sigma \left(v\right)=V_{1,1}-V_{1,0}V_{0,0}^{-1}V_{0,1},$$
where
$$V_{i,j}:=T^{-1}\sum_{t=1}^{T}v_{t-1+i}v_{t-1+j}^{⊤}.$$
In fact, Σ(v) is always (semi-)positive definite, as it takes the form of the Schur complement of the (semi-)positive-definite matrix $\left(\begin{array}{cc} V_{0,0} & V_{0,1}\\ V_{1,0} & V_{1,1} \end{array}\right)$. Thus, this maximum likelihood estimator readily satisfies condition (2).

### 6.3. Future Work

In this paper, we limit the VAR model to be bivariate for simplicity of analysis. We expect it is possible to generalize the current result to any higher dimensional VAR model. In such a generalization, feasible boundaries for the stable VAR models may require further effort to understand. 

## Figures and Tables

**Figure 1 entropy-23-00742-f001:**
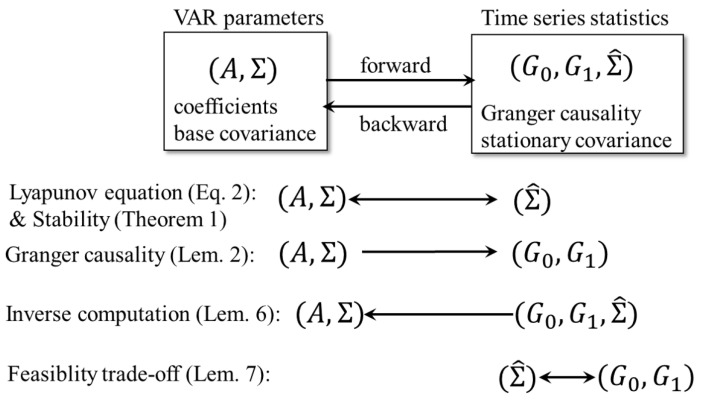
Schematic diagram of the organization of this paper.

**Figure 2 entropy-23-00742-f002:**
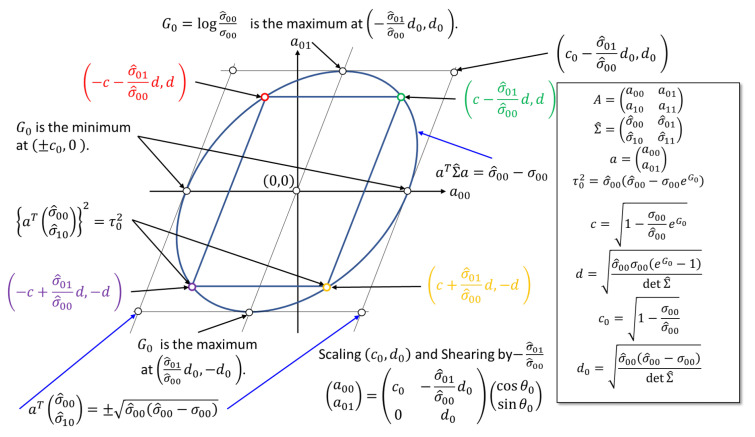
The ellipsis (Equation 42) and two parallel lines (Equation 45) (forming the parallelogram touching the ellipsis) on the plane $\left(a_{0,0},a_{0,1}\right)\in R^{2}$. The solution $\left(a_{0},a_{1}\right)$ is four intersections of these two (depicted by the colored points). Granger causality takes its maximum with the largest $|a_{0,1}|$ on the ellipsis and its minimum with $|a_{0,1}|=0$.

## Data Availability

Data sharing not applicable.

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
