# Peer review of "Designing Bivariate Auto-Regressive Timeseries with Controlled Granger Causality"

_entropy, 2021, doi:10.3390/e23060742_

Round 1
Reviewer 1 Report
The manuscript is clearly written and presented and addresses an important issue. The introduction includes some psychology as a motivation for this work. This isn't returned to at the end of the manuscript. As such, it may benefit from a few sentences about what relevance this work has to the field, given that it is a stated motivation and 5 out of the 9 references are from psychology.
minor
some numbers have not yet been added and have question mark place holders.
Author Response
(Please see the attachment.)
Dear our anonymous Reviewer 1,
Thank you for encouraging review comments on our manuscript. We revised our manuscript according to all the comments (no rebuttal). Please find our reply and revision in the attached file.
Sincerely,
Shohei Hidaka, Ph.D.
Takuma Torii, Ph.D.
Japan Advanced Institute of Science and Technology
1-1 Asahidai, Nomi, Ishikawa, 923-1292, Japan

Reviewer 2 Report
The authors present an exciting manuscript on how to generate bivariate time-series with given covariance and Granger causality. I have followed it with interest, and I can appreciate a rigorously work. Congratulations to the authors. However, I point out some comments to be addressed:
1 The introduction should highlight better the motivation of this type of model.
2 At the second part of the manuscript, several “??” should be replaced for the proper reference of proof or lemma.
3 A practical example should be shown so the readers from different areas can appreciate this model's usefulness.
Author Response
(Please see the attachment.)
Dear our anonymous Reviewer 2,
Thank you for encouraging review comments on our manuscript. We revised our manuscript according to all the comments (no rebuttal). Please find our reply and revision in the attached file.
Sincerely,
Shohei Hidaka, Ph.D.
Takuma Torii, Ph.D.
Japan Advanced Institute of Science and Technology
1-1 Asahidai, Nomi, Ishikawa, 923-1292, Japan
